# Mitochondrial MsrB2 serves as a switch and transducer for mitophagy

Seung Hee Lee[1,2,*] ID, Suho Lee[3], Jing Du[1], Kanika Jain[1], Min Ding[1], Anis J Kadado[1], Gourg Atteya[1], Zainab Jaji [1], Tarun Tyagi[1], Won-ho Kim[2], Raimund I Herzog[4], Amar Patel[5], Costin N Ionescu[6], Kathleen A Martin[1] & John Hwa[1,**] ID

## Abstract

Mitophagy can selectively remove damaged toxic mitochondria, protecting a cell from apoptosis. The molecular spatial–temporal mechanisms governing autophagosomal selection of reactive oxygen species (ROS)-damaged mitochondria, particularly in a platelet (no genomic DNA for transcriptional regulation), remain unclear. We now report that the mitochondrial matrix protein MsrB2 plays an important role in switching on mitophagy by reducing Parkin methionine oxidation (MetO), and transducing mitophagy through ubiquitination by Parkin and interacting with LC3. This biochemical signaling only occurs at damaged mitochondria where MsrB2 is released from the mitochondrial matrix. MsrB2 platelet-specific knockout and in vivo peptide inhibition of the MsrB2/LC3 interaction lead to reduced mitophagy and increased platelet apoptosis. Pathophysiological importance is highlighted in human subjects, where increased MsrB2 expression in diabetes mellitus leads to increased platelet mitophagy, and in platelets from Parkinson's disease patients, where reduced MsrB2 expression is associated with reduced mitophagy. Moreover, Parkin mutations at Met192 are associated with Parkinson's disease, highlighting the structural sensitivity at the Met192 position. Release of the enzyme MsrB2 from damaged mitochondria, initiating autophagosome formation, represents a novel regulatory mechanism for oxidative stress-induced mitophagy.

**Keywords** apoptosis; diabetes mellitus; methionine sulfoxide reductase; mitophagy; platelets
**Subject Categories** Metabolism; Neuroscience

See also: **A Kaur & EE Gardiner** (August 2019)

## Introduction

Currently, over 19.7 million adults in the USA have diagnosed DM and an estimated 8.2 million have undiagnosed DM (Benjamin *et al*, 2017). Diabetes mellitus (DM) is a progressive and chronic metabolic disorder characterized by hyperglycemia arising from impaired insulin levels, insulin sensitivity, and/or insulin action, leading to increased oxidative stress. Sixty-five percent of patients with DM will die from thrombotic events including heart attacks and strokes (Ferreiro & Angiolillo, 2011) where oxidative stress and platelets play a major role (Tang *et al*, 2011, 2014). Platelets are short-lived (7–10 days), small (2–4 μm), circulating anucleate cells, containing many critical factors required for regulation of thrombus formation, vascular homeostasis, and immune response (Lindemann *et al*, 2001; Leytin, 2012). Despite having no genomic DNA and thus limited transcriptional capabilities, platelets are capable of many fundamental cellular functions including *de novo* protein synthesis (Weyrich *et al*, 1998; Lindemann *et al*, 2001), programmed cell death (Mason *et al*, 2007), and autophagy (Feng *et al*, 2014; Ouseph *et al*, 2015; Lee *et al*, 2016). Platelets are prepackaged with the relevant mRNAs to allow for these well-orchestrated protective autophagy processes to maintain normal cellular function (basal autophagy) and to protect (induced autophagy) from severe oxidative stressors, as observed with diabetes mellitus or high-fat diet (Lee *et al*, 2016).

Autophagy and mitophagy play many emerging diverse critical roles in stemness and senescence (Palikaras *et al*, 2015; Garcia-Prat *et al*, 2016), progression to cell death in mitosis (Domenech *et al*, 2015), and resistance against infection (Manzanillo *et al*, 2013) through Parkin-dependent and Parkin-independent processes. Parkin, a ubiquitin E3 ligase, adds ubiquitin to many substrates leading to interactions with LC3, a key component of autophagosomes (Gegg *et al*, 2010; Geisler *et al*, 2010; Kane & Youle, 2011; Lee *et al*, 2016). Mitophagy is induced in human diabetic platelets through a PINK1- and Parkin-dependent process (Lee *et al*, 2016).

1   Yale Cardiovascular Research Center, Section of Cardiovascular Medicine, Department of Internal Medicine, Yale University School of Medicine, New Haven, CT, USA
2   Division of Cardiovascular Diseases, Center for Biomedical Sciences, National Institute of Health, Cheongju, Chungbuk, Korea
3   Departments of Neurology and Neurobiology, Cellular Neuroscience, Neurodegeneration and Repair Program, Yale University School of Medicine, New Haven, CT, USA
4   Section of Endocrinology, Department of Internal Medicine, Yale University School of Medicine, New Haven, CT, USA
5   Division of Movement Disorders, Departments of Neurology and Neurobiology, Yale University School of Medicine, New Haven, CT, USA
6   Yale Cardiovascular Medicine, Department of Internal Medicine, Yale-New Haven Hospital, New Haven, CT, USA
    *Corresponding author. Tel: +82 43 719 8664; E-mail: seunghee.lee216@gmail.com
    **Corresponding author. Tel: +1 203 737 5583; E-mail: john.hwa@yale.edu

The process of Parkin-dependent mitophagy is initiated by PINK1 accumulation in the OMM in response to mitochondrial depolarization and damage. PINK1 phosphorylates ubiquitin to activate the E3 ligase function of Parkin (Koyano *et al*, 2014) as well as the outer mitochondrial membrane protein MFN2 to serve as a Parkin receptor (Chen & Dorn, 2013). Once recruited, Parkin ubiquitinates multiple outer mitochondrial membrane proteins (e.g., VDAC1, HDAC6, and mitofusin) leading to interaction with LC3 and initiation of mitophagy (Shaid *et al*, 2013). In the high-oxidative stress DM environment, the molecular mechanism for selective removal of damaged mitochondria sparing adjacent intact mitochondria remains unclear.

We now report that the mitochondrial matrix protein methionine sulfoxide reductase B2 (MsrB2) is a Parkin substrate necessary for mitophagy induction. MsrB2 released from ruptured mitochondria reduces oxidized Parkin. In the absence of MsrB2, oxidized Parkin is inactive (aggregates). Whereas PINK1 accumulation identifies mitochondria undergoing oxidative stress, MsrB2–Parkin protein–protein interaction serves as a switch mechanism, allowing mitophagy to proceed only in mitochondria that are severely damaged or ruptured. This mechanism appears to occur in other nucleated cells.

# Results

Two mitochondria are residing side by side in the high-oxidative stress environment associated with diabetes mellitus: One is morphologically intact, while the other is severely damaged with loss of crista structure, swelling, and rupture (Fig 1A). A double membrane envelope (highlighted in red) appears to be forming at the site of loss of structural integrity. A key intriguing question is the molecular spatio-temporal mechanism that allows selection of the damaged mitochondria for autophagy, sparing the adjacent intact mitochondria. Such a mechanism may be critical for protection from apoptosis and improved cell survival.

## MsrB2 interacts with LC3 in DM platelets

We initially set out to determine unique requirements for the intriguing nucleus- and transcription-independent mitophagy process in DM platelets. As LC3 is a central component of autophagosome formation, we sought LC3-interacting proteins by inducing platelet mitophagy with CCCP, and immunoprecipitating LC3, followed by mass spectrometry (Appendix Fig S1A). We identified an unlikely interaction with methionine sulfoxide reductase B2 (MsrB2). MsrB2 reduces methionine sulfoxidation under conditions of severe oxidative stress (Fischer *et al*, 2012). However, MsrB2 resides in the mitochondrial matrix (Pascual *et al*, 2010; Ugarte *et al*, 2010; Fischer *et al*, 2012), and LC3 is a cytoplasmic membrane protein, suggesting our result may be an artifact of intense membrane solubilization, a common problem associated with immunoprecipitation. However, we noted multiple distinct LC3-interacting motifs (LIFs; Valdor & Macian, 2012) in both human MsrB2 (3 LIFs) and mouse MsrB2 (2 LIFs; Appendix Fig S1B). Moreover, MsrB2 was significantly induced in DM platelets, in contrast to MsrA and MsrB1 and MsrB3 (Appendix Fig S1C). Further exploration was warranted.

Given the potential pathophysiological relevance, we elected to determine native interactions in human DM patient platelets (non-DM served as the control), and rather than overexpressing proteins in cell culture systems, we performed the converse of the initial experiments, immunoprecipitating MsrB2. We detected an LC3 band (particularly LC3II, lower band) in diabetic human platelets (Fig 1B). To further support an MsrB2/LC3 interaction, we used high-resolution confocal microscopy. Individual platelets *in situ* demonstrated significant MsrB2 and LC3 colocalization only in DM platelets (Fig 1C). Moreover, immunoelectron microscopy demonstrated colocalization of small (5 nm; MsrB2) and large particles (15 nm; LC3) at mitochondria in DM platelets, with no significant colocalization in WT control (Fig 1D). Taken together, the selective increase in MsrB2 in DM platelets, the dual co-IP of LC3 and MsrB2, the LIFs on MsrB2, and the colocalization in platelets using confocal microscopy and immunoelectron microscopy all supported an MsrB2/LC3 interaction. MsrB2 may play a role in the mitophagy process.

## MsrB2 knockout leads to reduced mitophagy and increased platelet apoptosis

The interaction between MsrB2 and LC3 supports a role for MsrB2 in the regulation of mitophagy. MsrB2 knockdown (shMsrB2) significantly reduced mitophagy (LC3II, lower band) and mitophagy induced by $H_2O_2$ (Fig 2A). We additionally used a high-glucose (25 mM) stress, previously demonstrated to oxidatively stress platelets and induce a protective mitophagy response (Tang *et al*, 2014). Platelet mitophagy is recognized to protect against platelet apoptosis (Lee *et al*, 2016). Thus, with loss of mitophagy, we would expect a platelet apoptosis phenotype. Knockdown of MsrB2 leads to reduced mitophagy (LC3II, lower band) and significant increases in proapoptotic pp53(S15) (Fig 2B). Given our *ex vivo* immunoprecipitation results demonstrating an interaction between MsrB2 and LC3, and our *in vitro* knockdown results supporting a role for MsrB2 in mitophagy and thus preventing apoptosis, we then assessed for platelet apoptosis, *in vivo*. A global MsrB2 knockout mouse supported increased platelet apoptosis. An initial apoptosis array was followed by evidence of increased cytochrome c release (Appendix Fig S2A and B). Although these results support a role for MsrB2 in mitophagy, endothelial dysfunction (arising from MsrB2 knockout) may also lead to platelet apoptosis, and thus, we needed a platelet-selective knockout. Platelet-selective MsrB2 knockout (PF4 Cre MsrB2$^{-/-}$ compared to MsrB2 fl/fl, without the PF4 Cre) demonstrated increased reactive oxygen species (ROS) (Fig 3A) and significant loss of mitochondrial membrane potential (TMRE; Fig 3B and C). Increased platelet apoptosis was observed consistent with the reduced protective platelet mitophagy process (Fig 3D–F). However, as MsrB2 likely reduces MetO from many important mitochondrial proteins, to further demonstrate that the specific MsrB2 interaction with LC3 is important for mitophagy *in vivo*, we designed cell-penetrating peptides (based upon MsrB2 amino acid sequence; Appendix Fig S3A) to selectively disrupt the MsrB2/LC3-interacting motif (LIF) (Foroud *et al*, 2003). We included a Tat-control peptide (CP) (Appendix Fig S3A) as a negative control. Initial treatment of MEG-01 cells with a dose response and induction of stress with high glucose demonstrated reduced mitophagy and increased platelet apoptosis (pp53(S15) and active caspase-3) as compared to control peptide (Appendix Fig S3B). After peptide

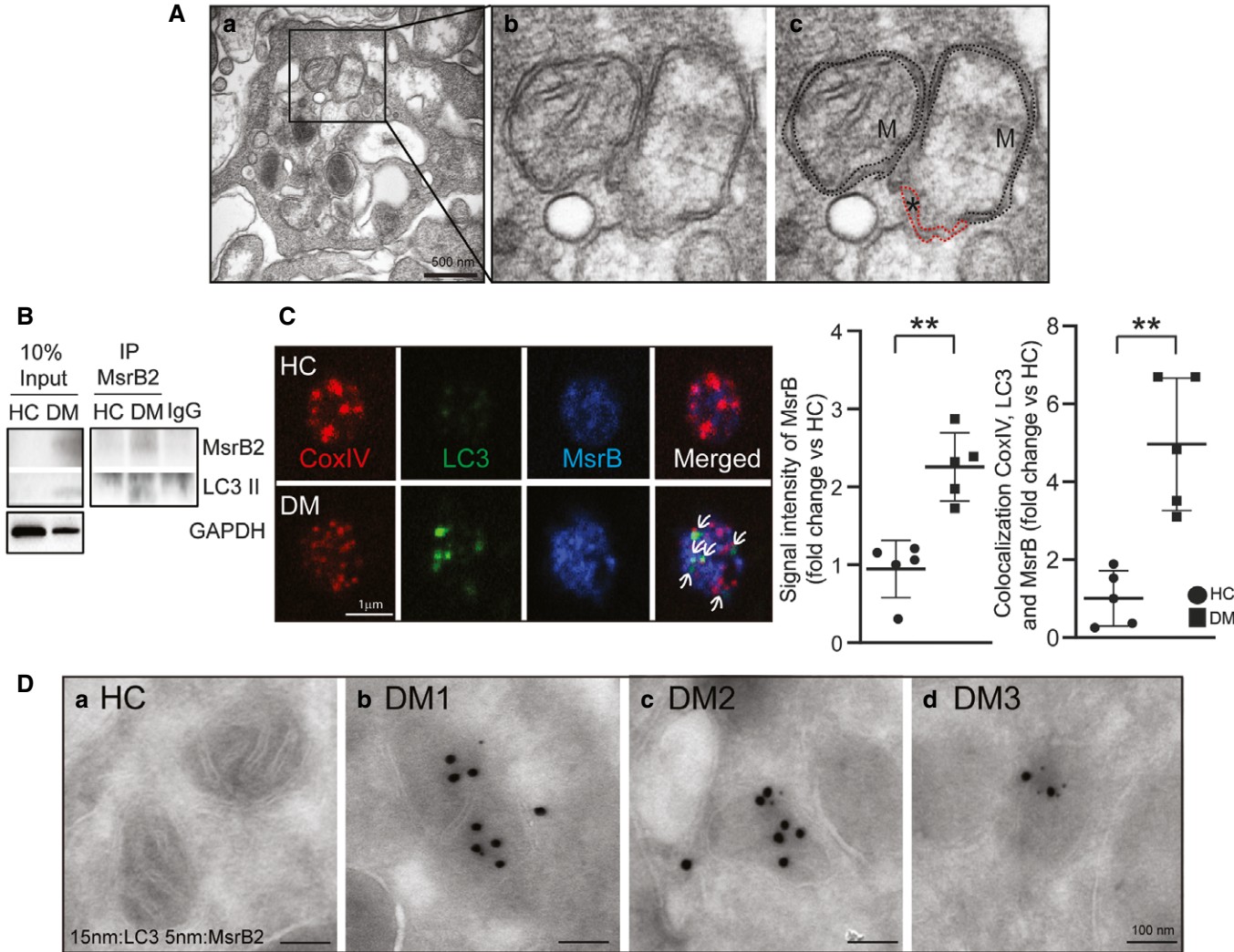

**Figure 1. Identification of MsrB2 as LC3 interaction protein in human platelets.**

A   Electron microscopy showing two adjacent mitochondria (one intact and one ruptured) in a DM platelet. a. Lower powered view of the single DM platelet. b. The enlarged inset highlights the ruptured mitochondria (with loss of crista morphology) next to the intact mitochondria. c. Black dashed line indicates mitochondrial membrane, and red dashed line indicates phagophore formation (M, mitochondria). * means loss of structural integrity of mitochondria membrane.

B   Immunoprecipitation (Meng *et al*, 2011) of MsrB2 in HC and DM human platelets followed by MsrB2 and LC3I/II Western blot analysis.

C   Confocal microscopy was used to corroborate the Western blot analysis using double staining for LC3 and MsrB2 in HC and DM platelets. Arrows indicate sites of colocalization of CoxIV, LC3, and MsrB2 as determined by the Volocity software (PerkinElmer, USA). Quantification is presented for the intensity of MsrB2 in HC vs DM and colocalization of CoxIV, LC3, and MsrB2 (MsrB2 intensity, **P = 0.0009; colocalization CoxIV and LC3, **P = 0.0014 vs. HC; n = 3). The nonparametric *t*-test was performed for comparisons of two groups. Analysis was performed with Prism software (GraphPad Software, Inc., La Jolla, CA). A difference of P < 0.05 was considered significant (mean ± SD).

D   LC3 and MsrB2 immuno-EM analysis of HC and DM platelets. 15-nm dots indicate immunogold-labeled LC3 clusters, and 5-nm dots indicate immunogold-labeled MsrB2 clusters. No significant clusters were found in HC (a) platelets. Representative areas of clusters of gold labeling in DM patients (b–d) are presented.

injection (LP or CP) into HFD mice (intraperitoneally for five consecutive days to induce mild/moderate oxidative stress), as anticipated Parkin, LC3I/II, and MsrB2 were all induced secondary to the increased oxidative stress from the HFD (Fig 3G and H). However, the mice treated with the inhibiting peptide (Foroud *et al*, 2003) (compared to HFD mice treated with control peptide) demonstrated significantly increased platelet apoptosis as assessed by active caspase-3 (Fig 3G and H). Taken together, both the cell culture and *in vivo* experiments support that MsrB2 interacts with LC3 and may be involved in removing mitochondria in preventing

platelet apoptosis. Given the *in vivo* phenotype, we proceeded to establish the molecular mechanism for this intriguing interaction between a mitochondrial matrix protein (MsrB2) and a cytoplasmic protein (LC3).

**MsrB2 also interacts with Parkin and prevents aggregation**

With upregulation of MsrB2 in DM platelets and interaction with LC3 (a key player in DM platelet mitophagy), and mitophagy being a Parkin-dependent process in diabetic platelets (Lee *et al*, 2016),

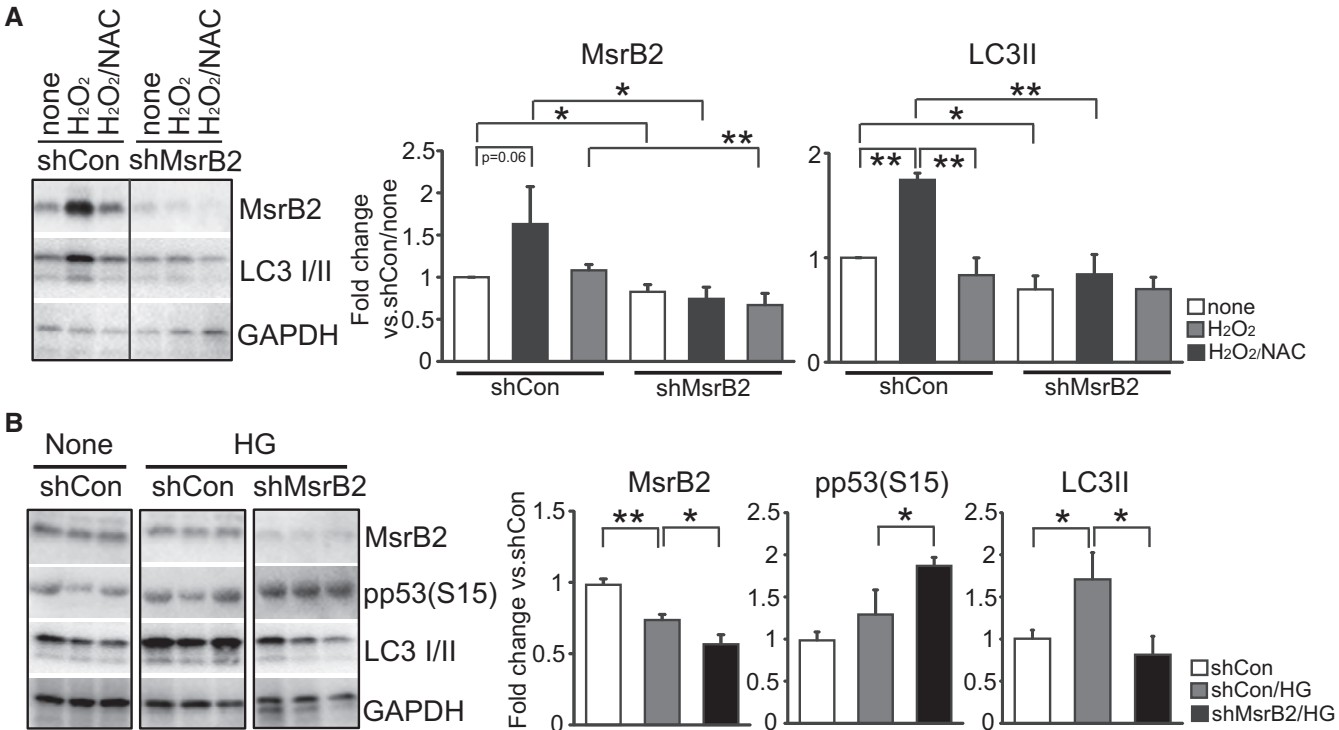

**Figure 2. MsrB2 is needed for mitophagy.**

A   Western blot analysis of MsrB2 and LC3I/II in MEG-01 cells after shMsrB2 transfection (72 h). Cells were then treated with $H_2O_2$ (1 mM for 1 h) alone or with NAC (100 μM for 30 min). GAPDH was used as the loading control. Quantification and analysis of individual groups. GAPDH served as the loading control (shMsrB2 vs. shCon, *$P$ = 0.024; shMsrB2/$H_2O_2$ vs. shCon/$H_2O_2$, *$P$ = 0.0295; shMsrB2/$H_2O_2$/NAC vs. shCon/$H_2O_2$/NAC, **$P$ = 0.0100; $n$ = 3, mean ± SD).

B   Western blot analysis of MsrB2, pp53(S15), and LC3I/II in MEG-01 cells after 25 mM HG treatment for 24 h. GAPDH was used as the loading control. Quantification analysis on individual groups. GAPDH served as the loading control (MsrB2: shCon/HG vs. shCon, **$P$ = 0.0017; shMsrB2/HG vs. shCon/HG, *$P$ = 0.0189; pp53: shMsrB2/HG vs. shCon/HG, *$P$ = 0.0317; LC3II: shCon/HG vs. shCon, *$P$ = 0.0215; shMsrB2/HG vs. shCon/HG, *$P$ = 0.0158; $n$ = 3, mean ± SD).

Data information: The nonparametric $t$-test was performed for comparisons of two groups. Analysis was performed with Prism software (GraphPad Software, Inc., La Jolla, CA). A difference of $P$ < 0.05 was considered significant.

Source data are available online for this figure.

we then sought to determine whether MsrB2 could also interact with Parkin. Immunoprecipitation of MsrB2 pulled down Parkin on Western blot analysis (Fig 4A). Conversely, immunoprecipitation of Parkin demonstrated a pulldown of MsrB2, particularly in DM platelets (Fig 4B). Confocal microscopy labeling Parkin, LC3, and MsrB2 on individual platelets demonstrated increased colocalization of all three components in DM versus control platelets (Fig 4C and D). This interaction between MsrB2 and key components of mitophagy strongly supports a complex functional relationship between MsrB2 and Parkin-dependent mitophagy. MetO on proteins serves as the substrate for methionine sulfoxide reductases (Msr; Ugarte et al, 2010; Fischer et al, 2012; Gu et al, 2015). Using a MetO antibody (in nonreducing conditions), oligomerized (high molecular weight) Parkin appears to have a MetO and thus may serve as a substrate for MsrB2 (Fig 4E). We then developed an in vitro assay using recombinant MsrB2 and assessing for the effect on Parkin aggregation (induced by $H_2O_2$) (Fig 4F). The last two lanes demonstrate that with little MsrB2 expression (red arrow), Parkin oligomerization is apparent. In contrast, substantial MsrB2 expression (blue arrow) prevents Parkin aggregation (Fig 4F). Taken together, the data suggest that in DM platelets, physical interaction can occur

between the mitochondrial matrix protein MsrB2 and the outer mitochondrial membrane Parkin. Parkin is MetO-modified and is associated with Parkin aggregation. This aggregation may be prevented by MsrB2 overexpression.

**Confirmation of MetO on Parkin**

Reactive oxygen species is significantly increased in DM platelets (Tang et al, 2014; Lee et al, 2016; Fig 5A). Protein modifications including 3-nitrotyrosine, 4-hydroxynonenal, carbonyl derivatives, and polyubiquitination are hallmarks of oxidative stress (Silva et al, 2015; Lee et al, 2016). Methionine sulfoxidation (oxidized methionine, MetO) is recognized to be one such modification leading to protein misfolding and aggregation (Stadtman, 2001; Moskovitz et al, 2016). Consistent with increased ROS, MetO-modified proteins are significantly increased by almost fourfold in diabetes mellitus (DM) platelets compared with healthy control (HC) platelets (HC: 1.1 ± 0.1, $n$ = 4; DM: 3.9 ± 0.3, $n$ = 12, $P$ < 0.01) (Fig 5B). The levels of MsrB2 and Parkin are increased in DM platelets (HC: 1.01 ± 0.03, $n$ = 4; DM: 1.97 ± 0.18, $n$ =12, $P$ < 0.01) (Fig 5B). Higher molecular weight Parkin bands are suggestive of protein

modification and aggregation in DM (Fig 5C and D). *In vitro* induction of ROS with $H_2O_2$ and rescue by the antioxidant NAC indicates that MsrB2 and Parkin levels are acutely regulated by oxidative

stress in platelets, which likely accounts for the increases seen in DM platelets (Appendix Fig S4A and B). The higher molecular weight band (120 kDa, aggregated) after Parkin was subjected to

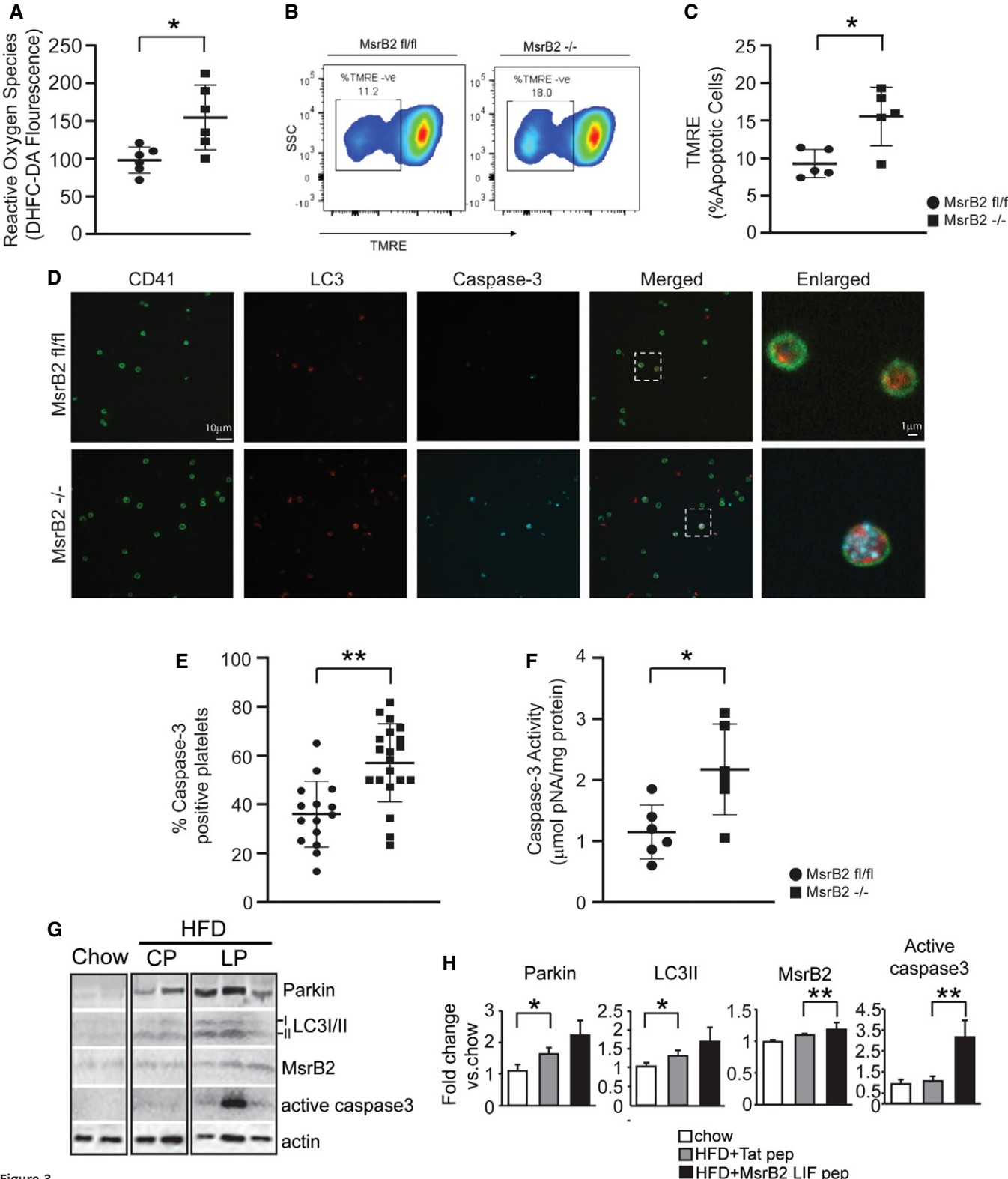

Figure 3.

**Figure 3. Platelet-specific deletion of MsrB2 leads to increased oxidative stress and intraplatelet apoptosis.**

A   Levels of reactive oxygen species (ROS) quantified in the platelets using the fluorescent dye DCFH-DA (ROS, *P = 0.0139; MsrB2 fl/fl n = 5, MsrB2$^{-/-}$ n = 6).

B, C   Representative image and quantification of mitochondrial apoptosis, measured in freshly isolated platelets from the MsrB2 knockout and age-matched floxed mice. Results are expressed as percentage of apoptotic cells (TMRE, *P = 0.0118; MsrB2 fl/fl n = 5, MsrB2$^{-/-}$ n = 5).

D   Representative images showing the intraplatelet expression of LC3 and active caspase-3 in CD41-stained platelets from the KO mice using fluorescent-labeled antibodies against the respective proteins of interest. Magnification: 100×. The rightmost panel represents zoom-in on an individual platelet showing the colocalization and expression of LC3 and active caspase-3.

E   Graph indicates platelets positive for active caspase-3 signal. The y-axis indicates percentage of caspase-3-positive cells from total cells calculated from n = 10 independent images per sample (active caspase-3 signal, **P = 0.0003).

F   Intracellular caspase-3 activity measured in freshly isolated platelets using the chromophore Ac-DEVD-pNA. Experiments performed in a minimum of n = 6 samples per group. Values expressed as mean ± SEM (caspase-3 activity, *P = 0.0157)

G, H   Western blot analysis of Parkin, LC3I/II, MsrB2, and active caspase-3 in platelets after cell penetration peptide injection in chow and HFD mice. Control peptide (CP), LIF peptide (Foroud et al, 2003). Quantification analysis of individual band intensity (Parkin, *P = 0.02964; LC3II, *P = 0.03200; MsrB2, **P = 0.0059; active caspase-3, *P = 0.0368 vs. chow or HFD group; each group n = 3). Actin served as the loading control.

Data information: The nonparametric t-test was performed for comparisons of two groups. Analysis was performed with Prism software (GraphPad Software, Inc., La Jolla, CA). A difference of P < 0.05 was considered significant. Mean ± SD (A, C, E and H).

$H_2O_2$ treatment was subjected to mass spectrometry to assess for protein modifications. Several peptide fragments were identified that contained a MetO (Met1, 80, 192, 327, 458) with no significant fragments in the control ($H_2O$-treated). An example of the raw data identifying Met192 is provided in Fig 5E. Mass spectrometry analysis was then performed on DM patient platelets and healthy control (HC) platelets to determine whether any of the identified MetO observed with $H_2O_2$ treatment was relevant in vivo. In DM platelets, we identified only MetO at Met192 (Fig 5E) with no significant modifications in the healthy control platelets. Supporting exquisite structural sensitivity of Met at this position, mutation of Met192 (M192L and M192V) on Parkin is recognized to be associated with Parkinson's disease (Meng et al, 2011). Taken together: (i) aggregation of Parkin is associated with MetO, (ii) MsrB2, which reduces MetO, interacts with Parkin, (iii) MsrB2 overexpression is able to reduce Parkin aggregation, (iv) the presence of human mutations at Met 192 is associated with Parkinson's disease (Meng et al, 2011), and (v) platelet Parkin knockout reduces mitophagy leading to increased platelet apoptosis and thrombosis (Lee et al, 2016), all suggest and support oxidative stress-induced MetO at position 192 on Parkin serving as a "brake" on platelet mitophagy.

### Parkin polyubiquitinates MsrB2 allowing interaction with LC3

As an E3 ubiquitin ligase, Parkin is recognized to play a key role in polyubiquitination of substrate proteins leading to interaction with LC3 and the mitophagy process. Polyubiquitination of proteins

under conditions of oxidative stress is characteristic of DM and can be partially reversed (in vitro) by reducing oxidative stress with NAC (Appendix Fig S4C). Western blotting for MsrB2 in DM platelets also reveals a "ladder" of higher molecular weight species, suggestive of ubiquitination (Appendix Fig S4D). Immunoprecipitating MsrB2 followed by Western blotting for ubiquitin confirmed that MsrB2 is indeed ubiquitinated (Fig 6A). Moreover, Parkin was also coimmunoprecipitated supporting a possible role for Parkin in ubiquitinating MsrB2 (Fig 6A). MsrB2 ubiquitination was also detected by immunostaining with ubiquitin and mitochondrial marker (CoxIV) signal colocalization (indicated by arrows) in DM platelets but not in healthy platelets (Fig 6B). To confirm MsrB2 polyubiquitination in a cell system, we transfected MsrB2-GFP and/or RFP-Parkin (with vector controls) in HEK293, followed by immunoprecipitation (anti-GFP-Trap bead) and Western blotting for ubiquitin. MsrB2 polyubiquitination was increased with cotransfection of Parkin (Fig 6C). Ubiquitination of MsrB2 was also enhanced by the mitophagy inducer CCCP in this system in the absence of overexpressed Parkin (Appendix Fig S4E). Taken together, these data demonstrate that MsrB2 reduces oxidized Parkin and, in turn, is ubiquitinated by Parkin, allowing it to interact with LC3 in the cytoplasm.

### MsrB2 is released from the mitochondrial matrix through mPTP opening and mitochondrial rupture

To provide insights into how Parkin and MsrB2 interact despite being in separate compartments, we isolated fresh mitochondria

**Figure 4. MsrB2 interacts with key mitophagy proteins, Parkin and LC3.**

A   Immunoprecipitation (Meng et al, 2011) of MsrB2 in HC and DM platelets, followed by Western blot analysis of precipitated Parkin and modified Parkin.

B   Immunoprecipitation (Meng et al, 2011) of Parkin in HC and DM platelets, followed by Western blot analysis of precipitated Parkin and MsrB2.

C   Confocal microscopy was used to corroborate the Western blot analysis using triple staining for Parkin, LC3, and MsrB2 in HC and DM platelets. Arrows indicate sites of colocalization of Parkin, LC3, and MsrB2 as determined by the Volocity software (PerkinElmer, USA). Graphical quantification of colocalization between Parkin, LC3, and MsrB2 signal.

D   Signal intensity of each group was converted to fold change and compared with the HC values (colocalization of Parkin, LC3, and MsrB2, **P = 0.0006). The nonparametric t-test was performed for comparisons of 2 groups. Analysis was performed with Prism software (GraphPad Software, Inc., La Jolla, CA). A difference of P < 0.05 was considered significant (mean ± SD, n = 3).

E   Immunoprecipitation (Meng et al, 2011) of Parkin in HC and DM platelets. Western blot analysis of MetO Parkin using specific Parkin and MetO antibodies. The detection of MetO Parkin was performed w/o β-mercaptoethanol (NonRe) SDS sample buffer.

F   In vitro assay. Parkin oxidation was induced with 1 mM $H_2O_2$ then incubated with MsrB2 protein at 37°C for 2 h. Western blot analysis of Parkin and MsrB2 was performed (red arrow, low MsrB2 expression; blue arrow, high MsrB2 expression).

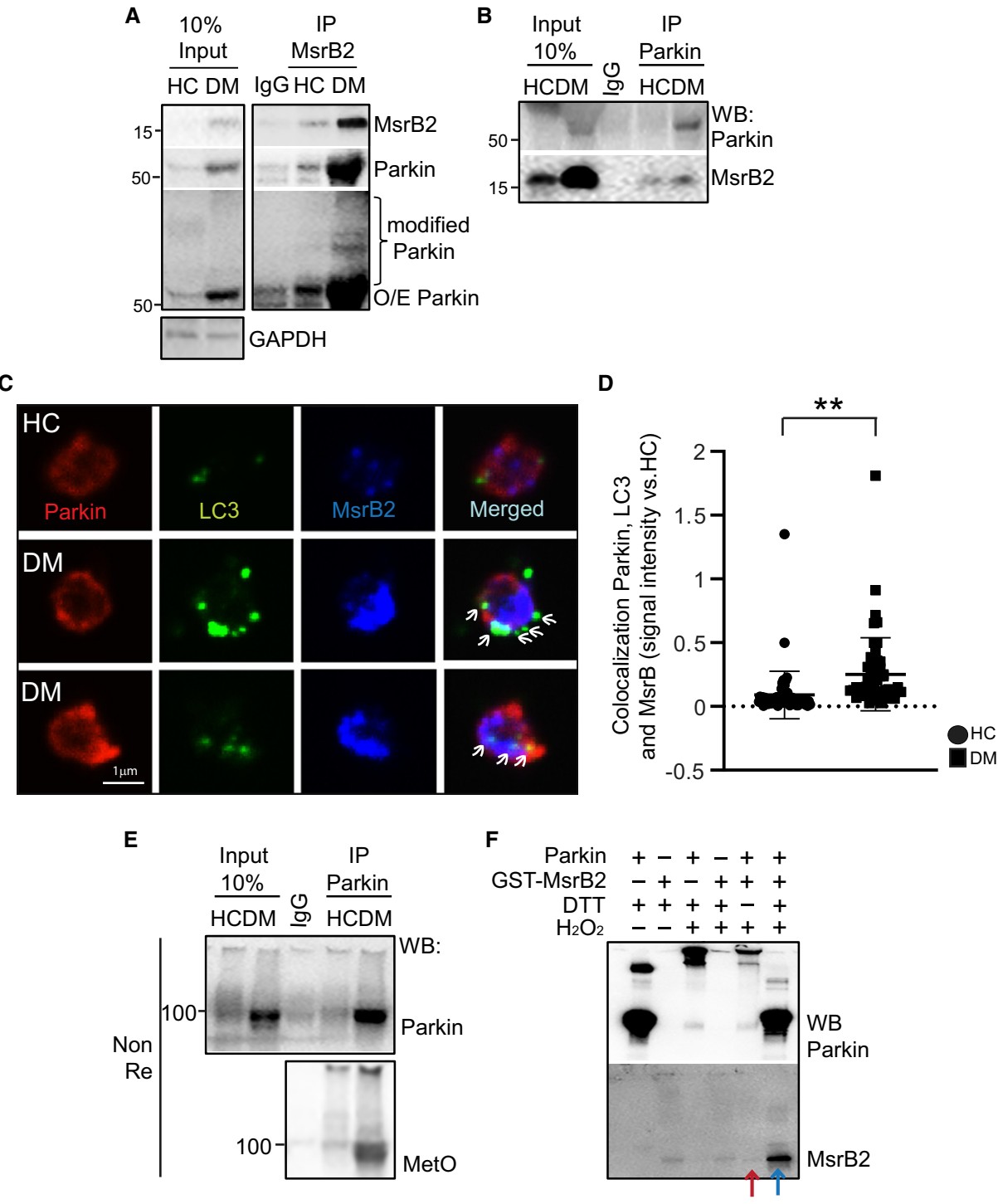

**Figure 4.**

from mouse heart tissue which is also known to be enriched in MsrB2 expression (Pascual *et al*, 2010). The abundance of mitochondria in the heart allows us to purify sufficient mitochondria to test the hypothesis that MsrB2 and Parkin come into contact when the mitochondrial membrane is leaky or disrupted. Initial opening of the mitochondrial permeability transition pore (mPTP) located in the inner mitochondrial membrane ultimately leads to

mitochondrial swelling and rupture (Halestrap, 2009). The mPTP inhibitor cyclosporine A (CsA) (Song *et al*, 2015) can be used to assess whether MsrB2 is released after mPTP opening. ATPB (a resident mitochondrial protein not known to traverse the mPTP) was used as a loading negative control, and cytochrome c, which moves from the outer aspect of the IMM to the cytosol through mechanical rupture of the OMM (Bax- and Bak-mediated), served as a further

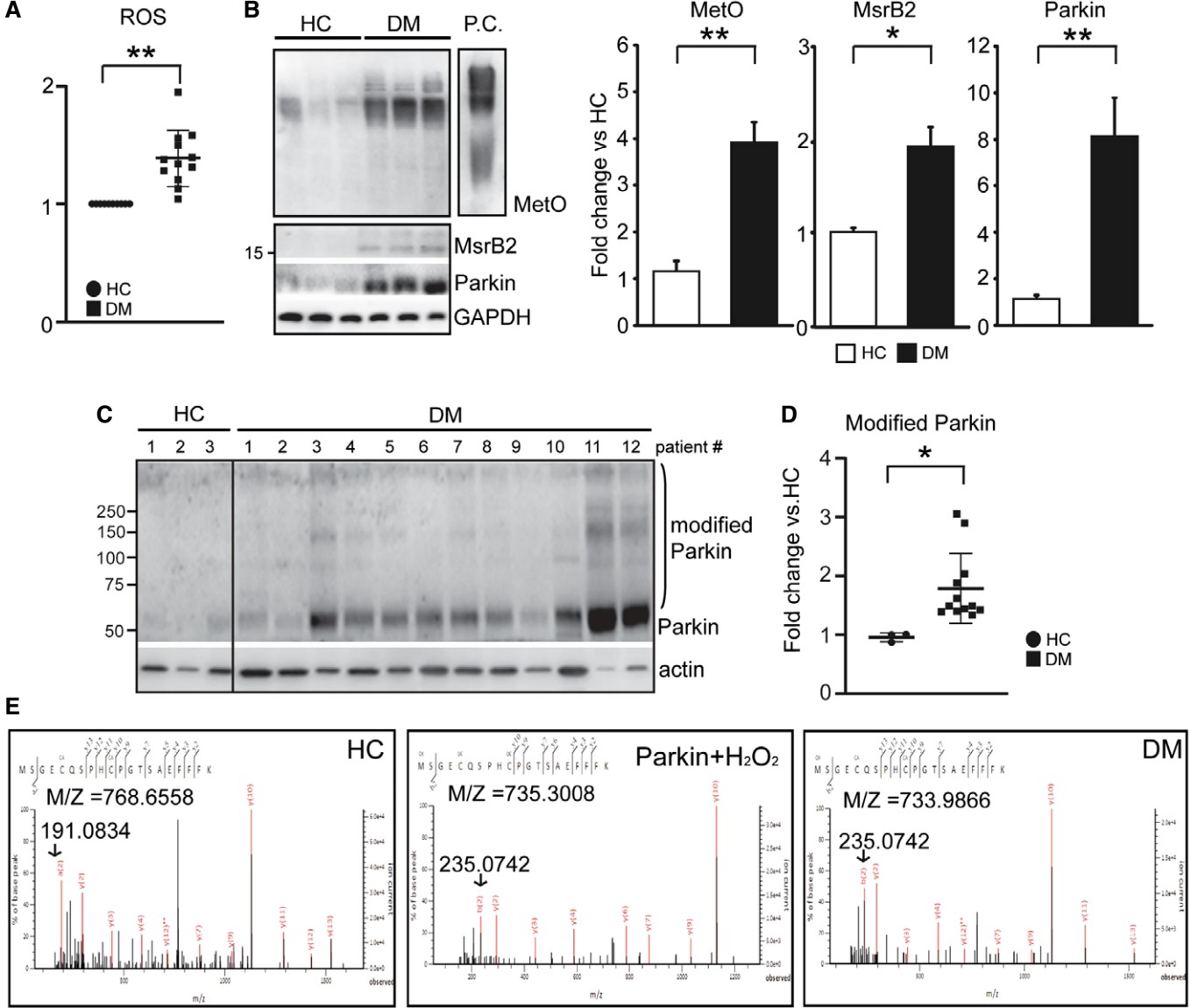

**Figure 5. Methionine sulfoxidation (MetO) and methionine sulfide reductase B2 (MsrB2) are increased, along with Parkin, in DM platelets.**

A   ROS levels were measured in platelet with FACS using specific oxidative stress detection dyes (CellROX; Molecular Probes, USA) in HC ($n = 8$) and DM ($n = 10$) platelets (ROS, **$P < 0.0001$ vs. HC).

B   Western blot analysis of methionine sulfoxidation (MetO), methionine sulfide reductase B2 (MsrB2), and Parkin in human healthy control (HC) and diabetic mellitus (DM) platelets. P.C. is a BSA-MetO positive control (Cayman Chemical Co.). Quantification of Western blot analysis shown in Fig 1C. Graphical representation and statistical analysis of HC ($n = 3$) and DM ($n = 12$) individuals (MetO, **$P = 0.0003$; MsrB2, *$P = 0.0186$; Parkin, **$P = 0.0003$ vs. HC).

C, D   Western blot analysis of Parkin and Parkin aggregation in human healthy control (HC) ($n = 3$) and diabetic mellitus (DM) ($n = 12$) platelets. Graphical representation and statistical analysis of HC ($n = 3$) and DM ($n = 12$) individuals (modified Parkin *$P = 0.035$ vs. HC). Actin served as the loading control.

E   Parkin IP was performed in HC, HC plus $H_2O_2$, and DM patients. MetO sites were analyzed by mass spectrometry using the Mascot program. Representative mass spectrometry analysis demonstrating identification of MetO192 peptide on Parkin.

Data information: The nonparametric $t$-test was performed for comparisons of two groups. Analysis was performed with Prism software (GraphPad Software, Inc., La Jolla, CA). A difference of $P < 0.05$ was considered significant. Mean ± SD (A, B and D).

Source data are available online for this figure.

control. In purified, isolated heart mitochondria, MsrB2 accumulated in the mitochondrial matrix with by CsA treatment as did cytochrome c (Fig 7A). We then used high-fat diet (HFD) to generate moderate levels of platelet oxidative stress in an *in vivo* mouse

model. Consistent with our human DM platelet studies, the oxidative stress associated with HFD feeding in mice leads to increased Parkin, MsrB2, and LC3II (lower band indicating active mitophagy) in non-DM mouse platelets (Fig 7B and C). Platelet staining with

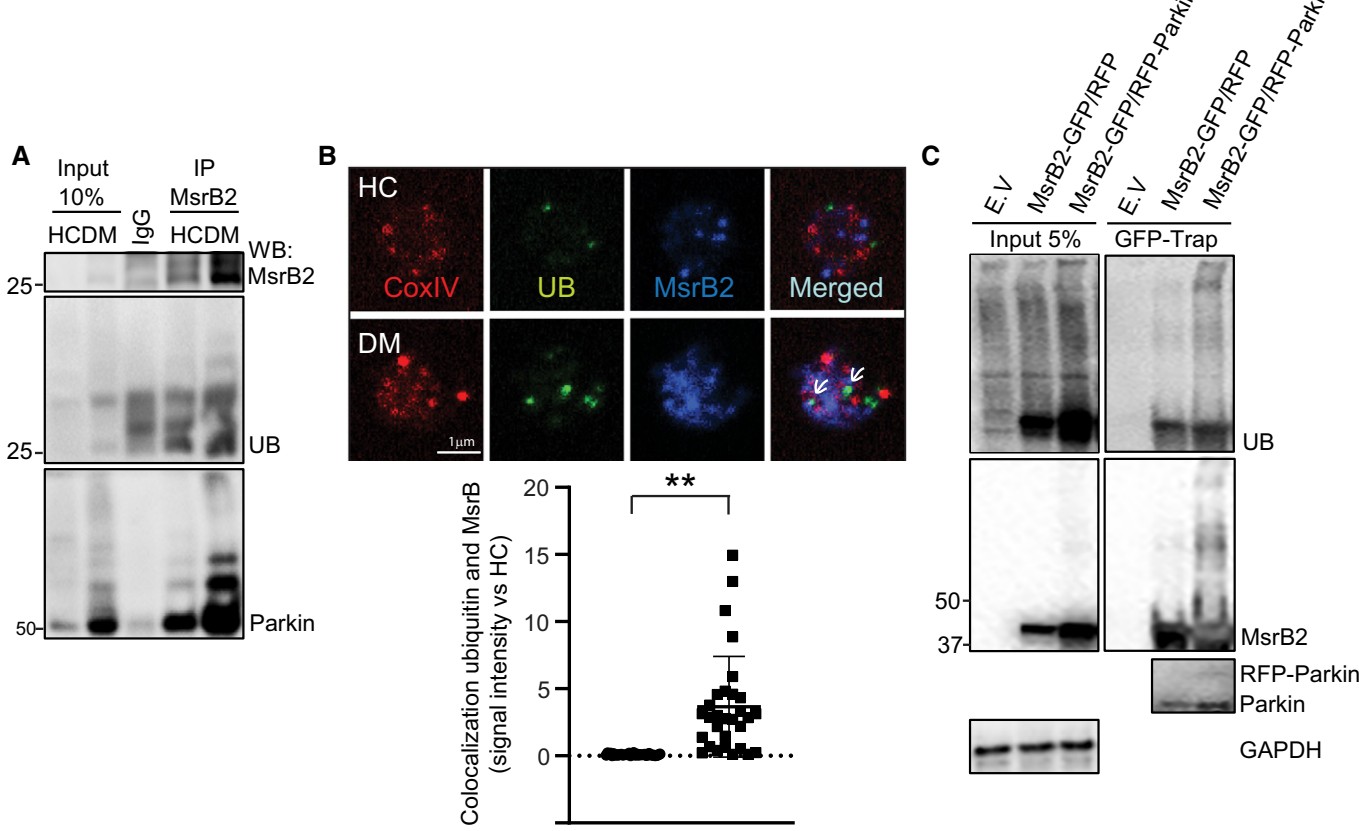

**Figure 6. MsrB2 is ubiquitinated by Parkin.**

A  Immunoprecipitation (Meng *et al*, 2011) of MsrB2 in HC and DM platelets. Western blot analysis of precipitated MsrB2, Parkin, and ubiquitinated MsrB2 (UB).
B  Confocal microscopy in individual platelets in HC vs DM assessing for CoxIV (mitochondrial marker), ubiquitin, and MsrB2. Colocalization is demonstrated by arrows and quantitated (colocalization ubiquitin, MsrB2, **$P$ = 0.0003 vs. HC). The nonparametric *t*-test was performed for comparisons of two groups. Analysis was performed with Prism software (GraphPad Software, Inc., La Jolla, CA). A difference of $P < 0.05$ was considered significant ($n$ = 3).
C  MsrB2 ubiquitination assay after transient transfection of MsrB2-GFP (1 μg) with RFP empty vector (E.V.) or RFP-Parkin (3 μg) in HEK293. After transfection (48 h), an IP was performed using GFP-Trap bead, followed by Western blot analysis using UB and MsrB2 and Parkin antibodies.

tubulin reveals overall shape changes indicative of stress responses in platelets from HFD mice (Fig 7D, panels a–b). Using ATPB as a mitochondrial marker, we confirmed colocalization (superimposition) of ATPB and MsrB2 in the mitochondrial matrix in platelets of chow-fed mice, with a staining pattern consistent with intact mitochondria (Fig 7Dc). Staining in platelets from HFD mice indicates disruption of the mitochondria and MsrB2 release from the mitochondrial matrix (loss of colocalization and wide distribution of MsrB2; blue in Fig 7Dd–e). We further confirmed MsrB2 mitochondrial release in readily transfectable H9c2 myoblasts, as MsrB2-GFP colocalized with CFP-Mito (mitochondrial marker) in transiently transfected cells (Appendix Fig S5). With CCCP treatment to induce mPTP opening and mitophagy, the MsrB2-GFP signal is more diffuse and no longer colocalizes with CFP-Mito, supporting stress-induced MsrB2 release from the mitochondrial matrix. These data collectively indicate that diverse cellular stresses promote mitochondria damage, resulting in MsrB2 release from the mitochondrial matrix to the cytosol, where it can interact with core autophagosome components, Parkin followed by LC3. In addition to platelets, we demonstrate that this important release process occurs with other cells, including mouse platelets, the nucleated mouse heart, and H9c2 cells.

## Parkinson's disease patients have reduced MsrB2 and reduced mitophagy

We have demonstrated that increased MsrB2 in DM platelets leads to increased mitophagy, serving as protection against apoptosis. Reduced MsrB2 expression (platelet-specific knockout) or interaction with LC3 (inhibiting LC3-interacting motif peptide) reduces mitophagy leading to increased apoptosis. We then sought the effects of reduced MsrB2 in human subjects. Parkin mutations are recognized to cause Parkinson's disease (Kitada *et al*, 1998); moreover, as described, mutation of Met192 (M192L and M192V) on Parkin (the same position as the MetO we detected in diabetes mellitus, MetO192) is also associated with Parkinson's disease (Foroud *et al*, 2003). Additionally, Msr enzymes have also been implicated in Parkinson's disease (Glaser *et al*, 2005). Based on these observations, we recruited consecutive consenting patients from the Parkinson's disease clinic at Yale to assess for MsrB2/mitophagy and

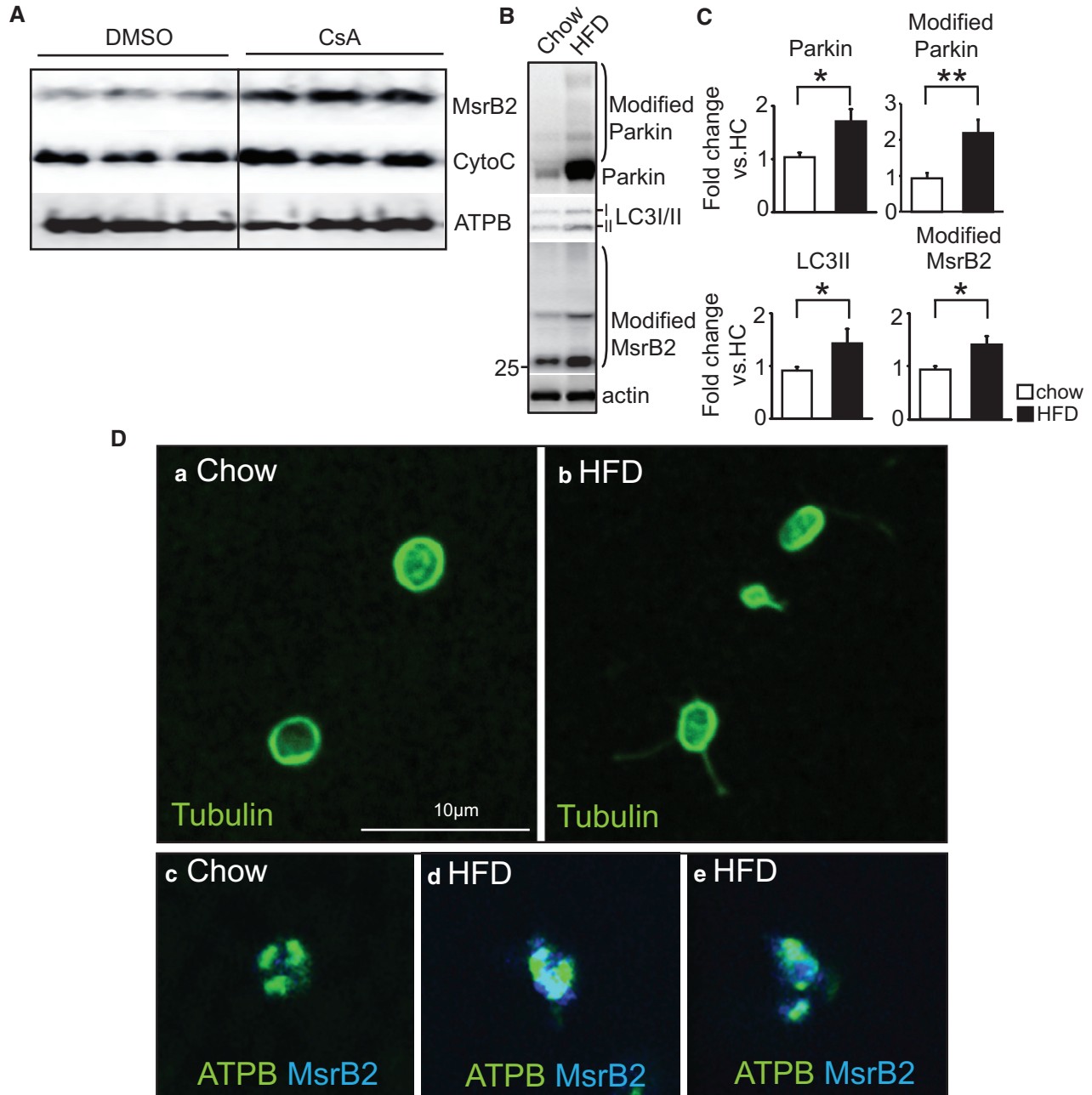

**Figure 7. MsrB2 is released by mitochondrial rupture.**

A   Western blot analysis of MsrB2 and cytochrome c in isolated mitochondria after incubation with or without cyclosporin A (CsA; 2 μM for 2 h). ATPB served as the loading and negative control (mitochondrial matrix protein).

B   Western blot analysis of Parkin, LC3I/II, and ubiquitinated MsrB2 in chow (n = 4) and HFD (n = 6) mouse platelets.

C   Quantification of individual band intensity (Parkin, *P = 0.0153; modified Parkin, **P = 0.0055; LC3II, *P = 0.0153; modified MsrB2, *P = 0.02245 vs. HC). Actin served as the loading control. The nonparametric t-test was performed for comparisons of two groups. Analysis was performed with Prism software (GraphPad Software, Inc., La Jolla, CA). A difference of P < 0.05 was considered significant (mean ± SD, HC n = 4, HFD n = 6).

D   Confocal microscopy was initially stained with tubulin antibody (image a. chow diet, or b. high-fat diet) to assess platelet architecture. This was followed by staining using ATPB (mitochondrial marker, green) and MsrB2 antibodies (Blue) (images c–e). There was loss of colocalization in the HFD mice supporting release (blue) from ruptured mitochondria (green) (images d and e).

platelet apoptosis. Although there was no significant difference in Parkin levels compared to age-matched controls, the Parkinson's disease patients demonstrated a significant reduction in MsrB2 with

reduction in LC3II and increased platelet apoptosis (Fig 8A and B). This is consistent with our DM studies and *in vivo* mouse studies showing an important link between reduced MsrB2, reduced

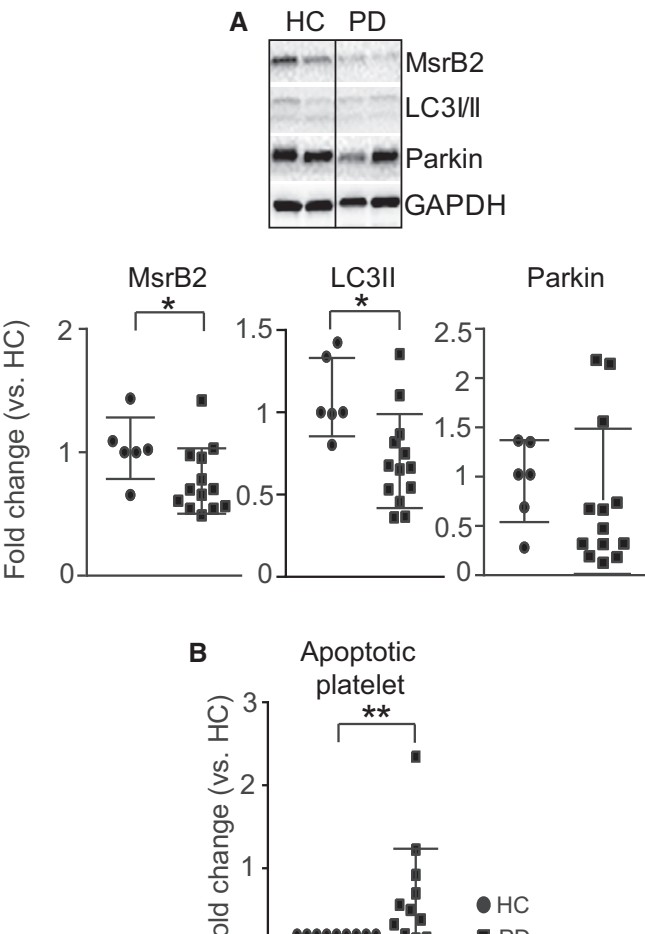

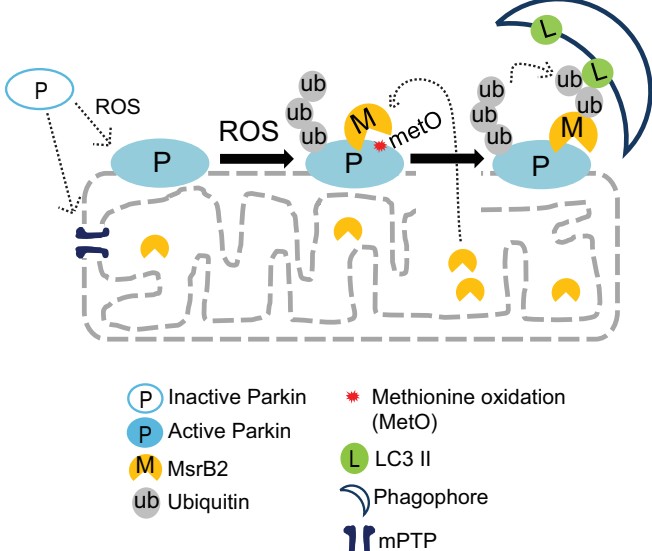

**Figure 9. Model for the molecular switch mechanism.**

A model outlining the key components required to switch on mitophagy in a high-oxidative (ROS) environment where Parkin has a MetO at position 192. Release of MsrB2 upon mitochondrial damage, swelling, and rupture removes the brake leading to formation of the autophagosome at the site of rupture. This is the underlying molecular process observed on the EM in Fig 1A, where mitophagy is taking place only with the ruptured mitochondria, sparing the adjacent intact mitochondria.

**Figure 8. Role of MsrB2 in Parkinson's disease platelets.**

A  Representative Western blot analysis of MsrB2, LC3I/II, and Parkin using Parkinson's disease (PD) platelets. Quantification analysis of individual band intensity (MsrB2, *$P = 0.0255$; LC3II, *$P = 0.0101$ vs. HC ($n = 5$) and PD ($n = 12$)). GAPDH served as the loading control (mean ± SD).

B  Platelet apoptosis (Annexin V) was measured by flow cytometry analysis in HC ($n = 9$) and PD ($n = 13$) platelets (**$P < 0.01$ vs. HC) (mean ± SD).

Data information: The nonparametric $t$-test was performed for comparisons of two groups. Analysis was performed with Prism software (GraphPad Software, Inc., La Jolla, CA). A difference of $P < 0.05$ was considered significant.

mitophagy, and increased platelet apoptosis. Further larger cohort studies are needed to explore whether MsrB2 can serve as a clinical biomarker for assessing disease development and progression.

**Model: A molecular switch mechanism for oxidative stress-induced mitophagy (Fig 9)**

High oxidative stress can lead to a vicious circle of ROS and mitochondrial damage. Parkin, a key protein in mitophagy, is translocated to the damaged mitochondria where in the presence of high ROS, methionine 192 is oxidized to methionine sulfoxide. Such modifications can lead to Parkin aggregation, preventing Parkin's ubiquitin E3 ligase activity. However, in the presence of mPTP

opening and mitochondrial swelling and rupture, MsrB2 is released from the mitochondrial matrix, reducing MetO on Parkin, allowing Parkin-mediated transfer of ubiquitin to MsrB2 and other proteins. The ubiquitinated MsrB2 then interacts with LC3 leading to formation of the double-membraned phagophore, surrounding the damaged mitochondria, and proceeding to mitophagy. Taken together, our studies, with support from the literature, provide a unique, transcriptionally independent mechanism (no genomic DNA involved) that selectively removes damaged toxic mitochondria (Fig 1A), protecting the cell from apoptosis.

## Discussion

Mitophagy can be generated by Parkin-independent or Parkin-dependent pathways (Gegg et al, 2010; Geisler et al, 2010; Shiba-Fukushima et al, 2012; Allen et al, 2013; Wauer et al, 2015). The process in platelets is Parkin-dependent and protects the platelet from oxidative stress- and mitochondrial-mediated damage (Lee et al, 2016). Accumulation of activated Parkin (arising from ROS-induced PINK1) in the OMM at damaged mitochondria ubiquitinates multiple outer mitochondrial membrane proteins leading to interaction with LC3, and initiation of mitophagy (Chen & Dorn, 2013; Koyano et al, 2014). We report that MsrB2 serves a dual role, both as a switch and as a transducer of Parkin-dependent mitophagy. Interestingly, being anucleate, this complex platelet process is independent of regulation from genomic DNA transcription.

## MsrB2 is a switch for mitophagy

Oxidative stress-induced methionine sulfoxidation is recognized to modify diverse proteins, altering function, and potentially leading to disease (Gu et al, 2015) [calmodulin (Gao et al, 1998), thrombomodulin (Glaser et al, 1992), CaMKII (Erickson et al, 2008)]. The methionine sulfoxide reductase family can reverse such protein damage (MsrA, B1, B2, and B3) (Marchetti et al, 2005; Cabreiro et al, 2008; Kwon et al, 2014), serving a protective role in all cells including blood cells (Rosen et al, 2009; Ouseph et al, 2015). Distinct from MsrA (mitochondria, cytosol, and nucleus), MsrB1 (cytosol and nucleus), and MsrB3 (ER and mitochondria), all localized to multiple cellular compartments, MsrB2 is located only in the mitochondrial matrix (Yermolaieva et al, 2004; Fischer et al, 2012). We report that MsrB2 plays a critical role in the process of oxidative stress-induced mitophagy, interacting with Parkin, removing MetO and thus serving as a switch, preventing aggregation, and allowing Parkin to ubiquitinate substrates. Our studies demonstrate a selective increase in MsrB2 in DM platelets suggesting that the upregulation of this mitochondria exclusive MetO reductant may serve an initial important role in protecting mitochondria in the high-oxidative stress environment observed with DM.

## MsrB2 is a transducer for mitophagy

Parkin is a well-known ubiquitin E3 ligase (Gegg et al, 2010; Kane & Youle, 2011; Cunningham et al, 2015) that is recognized to be post-translationally modified including N-nitrosylation (Chung et al, 2004), cysteine sulfonation (Meng et al, 2011), neddylation (Um et al, 2012), and methionine sulfoxidation (Meng et al, 2011). Under conditions of severe oxidative stress as with diabetes mellitus, Parkin MetO levels are increased. Mitochondrial damage and rupture result in release of mitochondrial matrix contents, permitting an important cooperative interaction between MsrB2 and Parkin, allowing Parkin to ubiquitinate MsrB2. Ubiquitinated MsrB2 (transducer) interacts with LC3, transducing the signal, leading to phagophore formation to surround and remove the damaged mitochondria. Thus, release of MsrB2 confers mitochondrial mitophagy specificity in allowing only those damaged/ruptured mitochondria to proceed with the phagophore formation. Based upon our human patient studies, this Parkin/MsrB2/LC3 protein complex may be an important principle for Parkin-dependent mitophagy in DM and other disease processes such as Parkinson's disease. This is the first description of such a switch/transducer mechanism, conferring damaged mitochondria selectivity.

### Clinical implications for MsrB2 regulation of mitophagy

MsrB2 thus serves an important role in the prevention of oxidative stress-induced platelet apoptosis in DM patients, reducing the recognized increased thrombosis associated with DM platelet apoptosis (Tang et al, 2014; Lee et al, 2016). However, cellular aging and neurodegeneration are also characterized by the accumulation of oxidized proteins and the reduced rate of repair or elimination of oxidized protein (Stadtman, 2001; Petropoulos & Friguet, 2006). Mitophagy clearly also plays an important fundamental role in these processes (Geisler et al, 2010; Bingol et al, 2014; Palikaras et al, 2015). Reduced MsrA and MsrB correlate with senile hair graying (Wood et al, 2009), epidermal damage (Schallreuter et al, 2006),

and vitiligo (Zhou et al, 2009). Msr have also been implicated with neurodegenerative diseases such as Alzheimer's and Parkinson's disease (Glaser et al, 2005), auditory function and hearing loss (Kwon et al, 2014; Kim et al, 2016), visual deterioration (Pascual et al, 2010), and obesity (Styskal et al, 2013), and insulin resistance (Kaneto et al, 2010). As we have highlighted, mutation of Met192 (M192L and M192V) is recognized to be associated with Parkinson's disease (Foroud et al, 2003), the same site as the MetO, and its reduction by MsrB2. This provides possible mechanistic insights linking MsrB2 to the development of neurodegenerative diseases. We now present provocative preliminary data from patients with Parkinson's disease where rather than increased MsrB2 and increased mitophagy as in DM, there is now reduced MsrB2 and reduced mitophagy, supporting our premise on the potential importance of MsrB2 in mitophagy and possibly importance in the process of aging and neurodegeneration. Further larger cohort studies are needed to confirm a direct link to neurodegeneration. Taken together, our studies support MsrB2 as being important not only for platelet protection from oxidative stress in DM, but likely protection from oxidative stress associated with aging and neurodegeneration. We provide a biomolecular mechanism as to the critical importance of MsrB2-induced mitophagy. Enhancing MsrB2 may be a novel treatment strategy for oxidative diseases such as DM and for Parkinson's disease.

# Materials and Methods

### Preparation of human platelets

Venous blood was drawn from healthy and diseased patients at Yale University School of Medicine (HIC#1005006865) from multiple outpatient clinics including the cardiovascular, diabetes, and neurology clinics. Informed consent was obtained from all subjects, and the experiments conformed to the principles set out in the WMA Declaration of Helsinki and the Department of Health and Human Services Belmont Report. All healthy subjects were free from medication or diseases known to interfere with platelet function (Tang et al, 2011, 2014; Lee et al, 2016) (Appendix Table S1). Upon informed consent, a venous blood sample (approximately 20 cc) was obtained by standard venipuncture and collected into tubes containing 3.8% trisodium citrate (w/v). Blood samples were prepared as previously described (Saxena et al, 1989). Platelet-rich plasma (PRP) was obtained by differential centrifugation. Purity of platelet preparation was determined by Western blot analysis using platelet markers (CD41), monocyte markers (CD14), and red blood cell markers (CD235a) (Lee et al, 2016).

### Preparation of mouse platelets

All mice were of C57Bl/6 background (WT, MsrB2 fl/fl, and MsrB2 whole-body or platelet-specific knockout). For the whole-body MsrB2 knockout studies, only 30-week-old females were used and compared to 30-week-old female wild-type mice. For the platelet-specific MsrB2 knockout studies, 8-week-old mice were fed for 12 weeks on high-fat diet. There were four male and two female knockout mice and three male and three female wild-type mice. All the animals were housed at the Yale Animal Facility

300 George St. New Haven, CT, under the supervision of Yale Animal Resources Center (YARC) and Rita Weber (Animal facility manager, Yale CVRC). All experiments were performed in accordance with guidelines and regulation as outlined by the Yale Institutional Animal Care and Use Committee (IACUC), under the approved protocol 2017-11413. For each animal study, the mice were consecutive bred mice. The mice were from the same genetic background and were often siblings, and thus, there was no significant variance within the groups. Any differences would therefore be directly related to treatment or modification. The experiments were also corroborated using other groups of mice. Where diabetes mellitus was induced, the mice were randomly assigned to induction. The experimenter was blinded to the level of blood glucose. As described in the Results section, further validation of experiments was performed using different approaches, i.e., chemical inhibition, chemical activation, and genetic knock-out. The combination of multiple randomized groups using multiple approaches was used to reduce any bias.

Blood (0.7–1 ml) was directly aspirated from the right cardiac ventricle into 1.8% sodium citrate (pH 7.4) in chow and HFD (high-fat diet for 12 weeks) (Yale IACUC #11413). For *in vivo* functional analysis of MsrB2, mice were treated intraperitoneally (i.p.) with the cell-penetrating peptide MsrB2 LIF (1 mg/kg) or control for 5 days (Zhang *et al*, 2016). Citrated blood from several mice of identical genotype was pooled and diluted with an equal volume of HEPES/Tyrode's buffer. PRP was prepared by centrifugation at 100 *g* for 10 min and then used for measuring platelet apoptosis and autophagy.

## Immunocytochemistry and confocal microscopy

Washed platelets ($1 \times 10^6$ cells/ml) were allowed to settle on glass-bottom dishes for 1 h prior to fixing with 4% paraformaldehyde solution (Santa Cruz Biotechnology). The platelets were then washed 2 × 5 min in PBS and permeabilized for 5 min in 0.25% Triton X-100/PBS. They were blocked for 60 min in 10% bovine serum albumin (Schallreuter *et al*, 2006)/PBS at 37°C and incubated in 3% BSA/PBS/primary antibody for 2 h at 37°C, or overnight at 4°C, and then washed 6 × 2 min in PBS, followed by an additional incubation for 45 min at 37°C in secondary antibody/3% BSA/PBS. Antibodies for MsrB2 (Abcam, USA; and laboratory made, USA), LC3 (Cosmo bio, Japan), Parkin (Abcam, USA), CoxIV (Santa Cruz, USA), tubulin (Sigma), and ATPB (Abcam) were used. The stained platelets were observed using a Nikon Eclipse-Ti confocal microscope with 100× oil immersion lens. Colocalization was assessed using parameters set in the Volocity software (PerkinElmer, USA). The colocalization was calculated within the signal area and the mean colocalization value compared with healthy controls or the non-treated group. The data are displayed as fold change, and statistical evaluation was performed as described below.

## Western blot analysis

Standard Western blot analysis protocols were used. Thirty micrograms of protein lysates was loaded in each well, and 3 or more independent replicates were used for quantification. We analyzed the band intensity using ImageJ analysis software (NIH) and converted the intensity value to fold change in comparison with

### The paper explained

#### Problem

Diabetes mellitus, leading to thrombovascular events, is a growing problem all over the world. Platelets play a key role in such pathological thrombotic events, largely from severe oxidative stress-induced damage. Mitophagy has recently been described as serving a protective role in preventing platelet dysfunction by removing damaged mitochondria. The molecular spatial–temporal mechanisms governing selective removal of damaged mitochondria remain unclear.

#### Results

We now report that the mitochondrial matrix protein MsrB2 plays an important role in switching on mitophagy by reducing Parkin methionine oxidation (MetO), and transducing mitophagy, through ubiquitination by Parkin and interacting with LC3. This switch occurs only at damaged mitochondria, as the mitochondrial matrix located MsrB2 requires release to the outer membrane, to switch on Parkin activity. MsrB2 platelet-specific knockout and *in vivo* peptide inhibition of the MsrB2/LC3 interaction confirmed that the reduced mitophagy increases platelet apoptosis, which leads to increased thrombosis. The importance of such a process was not only isolated to diabetic platelets but also observed in Parkinson's disease platelets where reduced MsrB2 expression was associated with reduced mitophagy. Release of MsrB2 from damaged mitochondria, initiating autophagosome formation, represents a novel regulatory mechanism for oxidative stress-induced mitophagy.

#### Impact

The elucidation of a mechanism for identifying and removing damaged mitochondria provides important insights into the pathophysiology of thrombotic events in diabetes mellitus. Moreover, new therapeutic targets in reducing heart and strokes in diabetes mellitus are provided.

HC or the non-treated group. Fold values were then used for statistical analysis. Antibodies and dilutions used are provided in Appendix Table S2.

## Immunoprecipitation

Healthy/DM platelet lysates and cell lysates (after transient transfection) were mixed with the specific target antibody (1 μg of LC3 anti-rabbit antibody (Abcam) and 2 μg of Parkin anti-goat antibody (Abcam) or 1.5 μg of Parkin anti-rabbit antibody (Abcam) for Parkin IP; 1.5 μg of MsrB2 anti-rabbit antibody (laboratory made) for MsrB2 IP; and 1 μg of GFP anti-rabbit antibody (Abcam, USA) or GFP-Trap bead (Chromotek, USA), and the same species IgG control with HC) and incubated overnight at 4°C. 50% slurry protein A sepharose bead and 50% slurry protein G sepharose bead were mixed 50:50. 30 μl of the 50% slurry washed A/G bead with lysates/antibody mixture was incubated for 1 h at 4°C. After three further washes with lysis buffer, we used 1–10% lysates for the input.

## Msr activity assays

The DDT-dependent assay was carried out using 500 ng of Parkin incubated with 5 mM DTT and 20 μl (5 μM) of recombinant GST-MsrB2 (Kim & Gladyshev, 2004). The reaction mixture was incubated at 4°C overnight and subjected to SDS–PAGE analysis.

## Measurement of reactive oxygen species

The oxidative stress levels were measured by CellROX (Molecular Probes; 1 µl CellROX (1:500)) using either human or mouse platelet (500 µl) incubated for 30 min at 37°C. Changes in fluorescence intensity were measured using flow cytometry (LSRII).

## *In vitro* ubiquitination

*In vitro* ubiquitination was performed using the Cayman assay per the manufacturer's protocol. We used recombinant Parkin (2.5 µM) and GST-MsrB2 (1 µM) followed by Western blot analysis.

## Mitochondrial isolation from mouse heart

Isolation of heart mitochondria was based upon a previously published protocol (Song *et al*, 2015). Briefly, mouse hearts freshly collected were minced and incubated with trypsin solution (in PBS) before homogenization with glass/Teflon Potter-Elvehjem homogenizer. Heart homogenates were centrifuged at 700 *g* for 10 mins at 400BAC and supernatant collected and centrifuged at 8,000 *g* for 10 min at 4°C. The pellet was washed and centrifuged at 8,000 *g* for 10 min at 4°C. The isolated mitochondrial suspension was treated with or without cyclosporin A (CsA; Sigma, USA) at room temperature, followed by Western blot analysis.

## Transient transfection

We purchased Parkin and MsrB2 ORF clones from OriGene (USA) and subcloned them into RFP- and GFP-tagged vectors, respectively. The MsrB2-GFP and pTagCFP-Mito (Evrogen, Russia) were introduced into HEK293 or H9c2 with Lipofectamine 3000 (Invitrogen, USA) according to the manufacturer's protocol. The shMsrB2 plasmid was introduced into MEG-01 cells with Nucleofector (Lonza, USA) according to the manufacturer's protocol. Empty vector and shCon were used for the control groups, respectively. Cells were harvested for 48 h or 72 h and lysed in lysis buffer (50 mM Tris–HCl (pH 7.4), 150 mM NaCl, 0.25% Triton X-100, and protease inhibitor cocktail) for further experiments.

## Statistics

Each experiment was carefully designed and analyzed with standard and accepted statistical analysis. Mouse studies were performed blinded to levels of glucose and/or genetic modification. We further validated results using independent complementary experiments as outlined in the Results section. Where appropriate, all data are expressed as mean ± SD or mean ± SE. A nonparametric *t*-test was performed for comparisons of two groups as outlined with the individual experiments. Analysis was performed with Prism software (GraphPad Software, Inc., La Jolla, CA). A difference of $P < 0.05$ was considered significant.

**Expanded View** for this article is available online.

## Acknowledgements

This work was funded by grants from NIH NHLBI (RO1 HL122815, HL115247, and HL117798). JD was supported by an NIH T32 Vascular Biology Training grant and ZJ by a Yale Hirsch and Gershon Fellowship.

## Author contributions

SHL wrote the manuscript, and planned and performed experiments, data analysis, and figure development; SL generated experimental materials and assisted with select experiments; JD, AJK, GA, and ZJ recruited patients for the studies and helped prepare samples; RIH, AP, and CNI supported the patient recruitment and experiments; KJ, MD, WK, and TT assisted with the experiment; KAM provided critical input for the studies and the manuscript; JH oversaw the study development and data analysis and wrote and edited the manuscript.

## Conflict of interest

The authors declare that they have no conflict of interest.

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
