## [Review Process File · EMBO Molecular Medicine]

Mitochondrial MsrB2 serves as a switch and transducer for mitophagy

Seung Hee Lee, Suho Lee, Jing Du, Kanika Jain, Min Ding, Anis J Kadado, Gourg Atteya, Zainab Jaji, Tarun Tyagi, Won-ho Kim, Raimund I Herzog, Amar Patel, Costin N. Ionescu, Kathleen A Martin and John Hwa.

Review timeline:

Submission date:	31 st January 2019
Editorial Decision:	25 th March 2019
Revision received:	3 rd May 2019
Editorial Decision:	6 th June 2019
Revision received:	7 th June 2019
Accept:	13 th June 2019

Editor: Celine Carret

Transaction Report:

1st Editorial Decision

25th March 2019

Thank you for the submission of your manuscript to EMBO Molecular Medicine. I apologise for the delay in reaching a decision. Although I was hoping to obtain a third evaluation, this referee is very late and never replied to our chasers. I am now proceeding based on the two consistent evaluations obtained so far as further delays cannot be justified.

You will see that the evaluations are positive and both referees have minor comments only that still need to be addressed in the next version of your article. Please note that depending on the nature of the revisions, this may be sent back to the referees for another round of review

REFeree REPORTS

Referee #1 (Comments on Novelty/Model System for Author):

Lee et al. dissect the role of the mitochondria matrix protein MsrB2 in regulating platelet mitophagy in high oxidant tone conditions such as diabetes. They identify MrsB2 by directional co-immunoprecipitation with LC3, a central protein in autophagosome formation, and colocalization studies in platelets from diabetic patients. These platelets had elevated levels of MrsB2 compared to healthy controls. Using multiple complimentary knockdown and knockout strategies in vitro and in vivo, the authors demonstrate that MrsB2 interacting with LC3 plays a functional role in mitophagy. They show that the E3 ubiquitin ligase parkin is an MsrB2 target for reduction of oxidized methionine in platelet mitochondria. They go on to demonstrate that MsrB2 in turn is ubiquitinated by parkin allowing it to interact with LC3. Importantly, they show that platelet mitochondria damage induced by oxidant stress in diabetes, results in MsrB2 release from the mitochondrial matrix to the cytosol, where it interacts with parkin and LC3 to induce mitophagy as a protective mechanism. Finally, they use platelets from Parkinson patients to illustrate the reverse case, the impact of reduction of MsrB2 function.

Referee #1 (Remarks for Author):

The manuscript is excellently written and provides important and novel insights into how mitophagy

protects platelets in high oxidant stress conditions, such as diabetes. The work shows very elegantly that MsrB2 acts as a switch to activate parkin and plays a direct role in mediating mitophagy. An intriguing translational implication is that MsrB2 may be targeted therapeutically to enhance protective mitophagy in oxidative stress diseases such as diabetes and or neurodegenerative diseases. I have no major criticism. The technical quality is outstanding.

Minor points:

- 1) Typo: first sentence of results: "...of cristae..."
- 2) Should it say "knockdown" instead of "knockout" in the results section? "...(Fig. 2B). Given our ex vivo immunoprecipitation results demonstrating an interaction between MsrB2 and LC3, and our in vitro knockout results supporting a role for MsrB2 in mitophagy and thus preventing apoptosis, we then assessed for platelet apoptosis, in vivo."
- 3) The mass spectrum of MetO (shown in figure 5E) is mislabeled as 4E in the results section.
- 4) Please explain the annotation of parent peptide M/Z and highlighted peaks in the mass spectra in the figure legend. Consider showing carbamidomethylated forms of the parent peptide and annotated b2 ions for all three conditions.

Referee #2 (Comments on Novelty/Model System for Author):

A new interaction is shown in multiple ways using different approaches, including platelets from healthy donors and patients as well as genetically modified mouse studies.

Referee #2 (Remarks for Author):

Autophagy contributes to the maintenance of intracellular homeostasis in a range of vascular cells including cardiomyocytes, endothelial cells, and arterial smooth muscle cells. Mitophagy is an autophagic response that specifically targets damaged cytotoxic mitochondria. In a high oxidative stress environment such as found in diabetes, how selective removal of a damaged mitochondria is achieved remains unclear. This clearly written manuscript from a group of researchers with expertise in this area describes the release of the enzyme methionine sulfoxide reductase (Msr) B2 from damaged mitochondria, initiating autophagosome formation. The overall conclusion is that MsrB2 can act on Parkin, reducing the oxidized (inactive) form of Parkin. An enormous amount of data is presented in the MS and supplementary files. The major strengths of the work are the novel findings and identification of MsrB2 as a molecular link between mitochondrial damage and induction of mitophagy and description of the interaction between MsrB2 and LC3 in platelets from patients with diabetes mellitus. This was shown in experiments using either MsrB2 or LC3 antibodies for IP followed by mass spectrometry; a role of MsrB2 to prevent aggregate formation of Parkin protein in the outer mitochondrial membrane is supported by data gained from both genetically modified mice and in vitro experiments. Authors have used platelets from both DM patients and patients with Parkinson's disease to demonstrate the importance of MsrB2 and that this pathway is important in pathophysiological contexts.

Specific points

How prevalent is the LC3 interacting motif in other proteins and across across biology? For example, the autophagy adaptor p62 (SQSTM1) also carries an LC3 binding domain. Do levels of p62 increase in DM platelets?

It would be of interest to evaluate MsrB2, LC3 and Parkin localization in DM platelets treated with mitoquinone or similar.

Given that levels of MsrB2 are elevated in diabetic platelets, can the authors be sure that the interaction is induced as a consequence of these altered levels. This might confound interpretation of experiments using confocal co-localization imaging.

The data is consistent with a novel regulatory mechanism for oxidative stress-induced mitophagy. Do other types of cellular stress also trigger the same pathway? What happens to the level of Parkin-mediated/mitochondrial ubiquitinylation in diabetic platelets? While multiple apoptotic markers and an apoptosis array were used, the authors have used only one

mitophagy marker LC3 to measure mitophagy triggered by changes in MsrB2 expression in DM platelets. Was mitochondrial clustering or then the role of MsrB2 in mitophagy would be more convincing.

The Authors have described the various statistical approaches that have been used in the study but it is not clear which statistical test has been used on which data set. It would be helpful to add this information to respective figure legends.

Other comments

The abstract contains a number of undefined abbreviations eg LO, ROS (an ROS?) Parkin MetO
Final sentence of introduction, "not only to be confined" is very confusing. Suggest replace with "This mechanism appears to occur in other nucleated cells".

The quality of the western blot in Fig 1B is not as good as later blots with this antibody. The authors could provide a densitometry analysis of the Figure 1B western blot to better differentiate between levels of LC3 in DM samples versus IgG control

Bar graphs showing the mean data of several replicates should always display the individual points e.g. 3E, 3A, 3C, 4D , 6B etc

|

We thank the Editor and the Reviewer's for the thorough and insightful review of the manuscript. We are particularly pleased that the mechanistic as well as the clinical relevance to disease processes were appreciated. We would like to present a point-by-point response to the remaining concerns raised by the Reviewers. All changes have been **highlighted** in the manuscript.

Referee #1

The manuscript is excellently written and provides important and novel insights into how mitophagy protects platelets in high oxidant stress conditions, such as diabetes. The work shows very elegantly that MsrB2 acts as a switch to activate parkin and plays a direct role in mediating mitophagy. An intriguing translational implication is that MsrB2 may be targeted therapeutically to enhance protective mitophagy in oxidative stress diseases such as diabetes and or neurodegenerative diseases. I have no major criticism. The technical quality is outstanding.

We thank the Reviewer for the insightful evaluation and for identifying important remaining concerns.

Minor points:

1) Typo: first sentence of results: "...of cristae..."

This has now been corrected

2) Should it say "knockdown" instead of "knockout" in the results section? "... (Fig. 2B). Given our ex vivo immunoprecipitation results demonstrating an interaction between MsrB2 and LC3, and our in vitro knockout results supporting a role for MsrB2 in mitophagy and thus preventing apoptosis, we then assessed for platelet apoptosis, in vivo."

We have replaced "knockout" with "knockdown".

3) The mass spectrum of MetO (shown in figure 5E) is mislabeled as 4E in the results section.

This has now been corrected.

4) Please explain the annotation of parent peptide M/Z and highlighted peaks in the mass spectra in the figure legend. Consider showing carbamidomethylated forms of the parent peptide and annotated b2 ions for all three conditions.

Thanks for highlighting this oversight on our part. The explanation and forms have now been provided. Representative mass spectrometry analysis demonstrating a healthy control (HC) Parkin peptide (containing Met192) of 769 (M/Z, mass/charge), and oxidative stressed peptide of 735 (M/Z) and DM peptide of 734 (M/Z). Fragmentation identified an a(2) of 191 in healthy control (HC) subjects. The b(2) fragment identified (in H₂O₂ treated and DM patients) has the a(2) fragment (mw 191) plus C=O (mw 12+16) plus the O on Met192 (mw 16) giving a final molecular weight of 235. This has now been added to the figure legend.

Referee #2:

Autophagy contributes to the maintenance of intracellular homeostasis in a range of vascular cells including cardiomyocytes, endothelial cells, and arterial smooth muscle cells. Mitophagy is an autophagic response that specifically targets damaged cytotoxic mitochondria. In a high oxidative stress environment such as found in diabetes, how selective removal of a damaged mitochondria is achieved remains unclear. This clearly written manuscript from a group of researchers with expertise in this area describes the release of the enzyme methionine sulfoxide reductase (Msr) B2 from damaged mitochondria, initiating autophagosome formation. The overall conclusion is that MsrB2 can act on Parkin, reducing the oxidized (inactive) form of Parkin. An enormous amount of data is presented in the MS and supplementary files. The major strengths of the work are the novel findings and identification of MsrB2 as a molecular link between mitochondrial damage and induction of mitophagy and description of the interaction between MsrB2 and LC3 in platelets from patients with diabetes mellitus. This was shown in experiments using either MsrB2 or LC3 antibodies for IP followed by mass spectrometry; a role of MsrB2 to prevent aggregate formation of Parkin protein in the outer mitochondrial membrane is supported by data gained from both genetically modified mice and in vitro experiments. Authors have used platelets from both DM patients and patients with Parkinson's disease to demonstrate the importance of MsrB2 and that this pathway is important in pathophysiological contexts.

We thank the Reviewer for the thorough review and the important remaining concerns.

Specific points.

1) How prevalent is the LC3 interacting motif in other proteins and across biology? For example, the autophagy adaptor p62 (SQSTM1) also carries an LC3 binding domain. Do levels of p62 increase in DM platelets?

This is an important point as highlighted by the Reviewer. LC3 interacting motifs (LIRs) have a number of configurations.

LC3 interacting motifs (LIRs)		
W		L
Y	X1X2	V
F		I

LIRs can be found in diverse proteins¹⁻⁷. Meticulous evaluation is needed to confirm the motif to be relevant and real. We have checked p62 in DM platelets and observed no significant change. The explanation may be differences in platelet autophagy response to stress compared to nucleated cells. We have previously observed differences in the mitophagy machinery in platelets⁸.

2) It would be of interest to evaluate MsrB2, LC3 and Parkin localization in DM platelets treated with mitoquinone or similar.

This is a great suggestion. We have used N-acetylcysteine (NAC) for our experiments and demonstrated a reduction in oxidative stress induced mitophagy (**Fig. 2A, Appendix Fig. 4A, 4B and 4C**). We now further demonstrate that Parkin, LC3 and MsrB2 colocalization (**Reviewer's response Figure 1A &1B**) as well as ubiquitin, Parkin and MsrB2 colocalization (**Reviewer's response Figure 1C &1D**) are reduced in NAC treated platelets.

3) Given that levels of MsrB2 are elevated in diabetic platelets, can the authors be sure that the interaction is induced as a consequence of these altered levels. This might confound interpretation of experiments using confocal co-localization imaging.

The Reviewer has raised an excellent point. We provide evidence for MsrB2 induction in diabetic platelets and confirm increased colocalization among MsrB2, Parkin and LC3 (**Fig 4C and D**). Moreover, we confirmed that interaction between Parkin and MsrB2 through IP experiments (**Fig 4A and B**). For the healthy control where MsrB2 is not induced, reduction of MsrB2 with shMsrB2 (**Figure 2**), platelet selective MsrB2 knockout mice (**Figure 3**), and Parkinson's disease patients (**Figure 8**) there appears to be reduced mitophagy and enhanced apoptosis. Multiple approaches were needed to answer this important concern which we also had.

4) The data is consistent with a novel regulatory mechanism for oxidative stress-induced mitophagy. Do other types of cellular stress also trigger the same pathway?

We have been exploring other cells including cardiomyocytes, particularly myoblast (H9C2 cells) (**Appendix Fig 5**). Active LC3 (LC3II) is increased in MsrB2 overexpression in H9C2 cells (**Reviewer's response figure 2A**). We additionally confirm an interaction between MsrB2, Parkin and LC3 using HEK293 cells (**Reviewer's response figure 2B and 2C**). Moreover, endothelin-1 (ET-1, cardiac damage inducer) increased MsrB2 expression and LC3 activation (**Reviewer's response figure 2D**). Functional studies are currently being performed on cardiomyocytes and H9C2 cells. Our data for Parkinson's disease (**Figure 8**) also suggests that such a process (MsrB2 regulation of Parkin mediated mitophagy) may also be important for the central nervous system. We therefore believe that this process is likely to be found in many cells.

5) What happens to the level of Parkin-mediated/mitochondrial ubiquitylation in diabetic platelets?

This was also an important question that we had to address. Parkin mediated ubiquitination of MsrB2 is increased in diabetic platelets following MsrB2 induction (**Figure 6A**). Reduction of oxidized Parkin recovers Parkin's function as a ubiquitin E3 ligase.

6) While multiple apoptotic markers and an apoptosis array were used, the authors have used only one mitophagy marker LC3 to measure mitophagy triggered by changes in MsrB2 expression in DM platelets. Was mitochondrial clustering or then the role of MsrB2 in mitophagy would be more convincing.

We have confirmed the induction of several autophagy components in a previous reports⁸. Beclin1, ATG3, ATG7 and ATG12-5 complex were all increased in DM patients (**Reviewer's response Figure 3**⁸). Mitophagy related protein (PINK1 and Parkin) were also increased in DM platelets (**Reviewer's response Figure 3**). With MsrB2 overexpression, there is increased LC3 II (active form of LC3, lipidated form).

7) The Authors have described the various statistical approaches that have been used in the study but it is not clear which statistical test has been used on which data set. It would be helpful to add this information to respective figure legends.

We have now added the statistical tests to the figures.

Other

comments

1) The abstract contains a number of undefined abbreviations eg LO, ROS (an ROS?) Parkin MetO

We have now defined the abbreviations in the abstract.

2) Final sentence of introduction, "not only to be confined" is very confusing. Suggest replace with "This mechanism appears to occur in other nucleated cells".

The sentence has now been changed.

3) The quality of the western blot in Fig 1B is not as good as later blots with this antibody. The authors could provide a densitometry analysis of the Figure 1B western blot to better differentiate between levels of LC3 in DM samples versus IgG control.

We have now provided an improved figure. This is an immunoprecipitation for a native protein interaction in platelets, and thus the signal was relatively weak compared to overexpression IPs. Multiple other experiments were thus necessary to confirm this interaction (**Fig 1C and Fig 1D**).

4) Bar graphs showing the mean data of several replicates should always display the individual points e.g. 3E, 3A, 3C, 4, 6B etc.

We have now changed the figure formats.

REFERENCES

1. Birgisdottir AB, Lamark T and Johansen T. The LIR motif - crucial for selective autophagy. *J Cell Sci.* 2013;126:3237-47.
2. Cheng X, Wang Y, Gong Y, Li F, Guo Y, Hu S, Liu J and Pan L. Structural basis of FYCO1 and MAP1LC3A interaction reveals a novel binding mode for Atg8-family proteins. *Autophagy.* 2016;12:1330-9.
3. Hubbard VM, Valdor R, Macian F and Cuervo AM. Selective autophagy in the maintenance of cellular homeostasis in aging organisms. *Biogerontology.* 2012;13:21-35.
4. Kirkin V, Lamark T, Sou YS, Bjorkoy G, Nunn JL, Bruun JA, Shvets E, McEwan DG, Clausen TH, Wild P, Bilusic I, Theurillat JP, Overvatn A, Ishii T, Elazar Z, Komatsu M, Dikic I and Johansen T. A role for NBR1 in autophagosomal degradation of ubiquitinated substrates. *Mol Cell.* 2009;33:505-16.
5. Liu X, Li Y, Wang X, Xing R, Liu K, Gan Q, Tang C, Gao Z, Jian Y, Luo S, Guo W and Yang C. The BEACH-containing protein WDR81 coordinates p62 and LC3C to promote autophagy. *J Cell Biol.* 2017;216:1301-1320.
6. Rogov VV, Stolz A, Ravichandran AC, Rios-Szwed DO, Suzuki H, Kniss A, Lohr F, Wakatsuki S, Dotsch V, Dikic I, Dobson RC and McEwan DG. Structural and functional analysis of the GABARAP interaction motif (GIM). *EMBO Rep.* 2017;18:1382-1396.
7. Seillier M, Peugot S, Gayet O, Gauthier C, N'Guessan P, Monte M, Carrier A, Iovanna JL and Duseti NJ. TP53INP1, a tumor suppressor, interacts with LC3 and ATG8-family proteins through the LC3-interacting region (LIR) and promotes autophagy-dependent cell death. *Cell Death Differ.* 2012;19:1525-35.
8. Lee SH, Du J, Stitham J, Atteya G, Lee S, Xiang Y, Wang D, Jin Y, Leslie KL, Spollett G, Srivastava A, Mannam P, Ostriker A, Martin KA, Tang WH and Hwa J. Inducing mitophagy in diabetic platelets protects against severe oxidative stress. *EMBO Mol Med.* 2016;8:779-95.

Thank you for the submission of your revised manuscript to EMBO Molecular Medicine. We have now received the enclosed reports from the referee who was asked to re-assess it. As you will see the reviewer is now supportive and I am pleased to inform you that we will be able to accept your manuscript pending editorial final amendments.

REFeree REPORTS.

Referee #2 (Comments on Novelty/Model System for Author):

stated in previous review

Referee #2 (Remarks for Author):

Thank you for the revision and discussion of my points. Highly interesting paper.

Corresponding Author Name: John Hwa, Seung Hee Lee
 Journal Submitted to: EMBO Molecular Medicine
 Manuscript Number: EMM-2019-10409